# A new strategy for durable control of late blight in potato by a single soil application of an oxathiapiprolin mixture in early season

Yigal Cohen *, Avia E. Rubin

Faculty of Life Sciences, Bar Ilan University, Ramat Gan, Israel

* ycohen@biu.ac.il, yigal.cohen1@gmail.com

**Data Availability Statement:** All data are included in the manuscript.

**Funding:** The author(s) received no specific funding for this work.

## Abstract

Root treatment with oxathiapiprolin, benthiavalicarb or their mixture Zorvec-Endavia [ZE (3 +7, w/w)] was shown to provide prolonged systemic protection against foliar oomycete pathogens attacking cucumber, tomato and basil. Here we report that these fungicides can effectively protect potato plants against late blight when applied to the soil in which such potato plants are grown. In two field experiments, performed in 2019 and 2020, potato plants grown in 64 L containers were treated with a soil drench of oxathiapiprolin, benthiavalicarb or ZE at 12.5, 25 or 50 mg ai/five plants in a container. Artificial inoculations with *Phytophthora infestans* revealed that such treated plants were protected against late blight in a dose-dependent manner all along the season. Interestingly, oxathiapiprolin persisted in the treated soil for at least 139 days, providing systemic protection against late blight to the following potato crops grown in that treated soils. Potato plants grown in loess soil in the field were either sprayed or drenched with ZE. Plants treated via the soil were significantly better protected against late blight compared to the plants treated by a spray. The data demonstrate a new strategy for season-long protection of potato against late blight by a single soil application of ZE. The systemic nature of oxathiapiprolin and benthiavalicarb composing ZE assures the translocation to the foliage of two fungicides with different modes of action. This shall minimize the risk of developing resistance against either fungicide in the treated crops.

## Introduction

Late blight caused by the oomycete *Phytophthora infestans* (Mont.) De Barry is a devastating disease of potato and tomato world-wide. Fungicidal sprays serve as a major measure for combating the disease. Thirty-six fungicides and fungicidal mixtures are currently (March 2020) registered in Europe for late blight control (www.EuroBlight) including contact, translaminar and systemic products. The recently introduced fungicides are oxathiapiprolin (2017), oxathiapiprolin+famoxadone (2018), oxathiapiprolin+amisulbrom (2018), benthiavalicarb (2018) and oxathiapiprolin+benthiavalicarb (2019). Oxathiapiprolin (OXPT) is a new piperidinyl thiazole isoxazoline fungicide (FRAC code U15) that targets the oxysterol binding proteins in oomycete cells [1]. It is extremely active against plant pathogenic oomycetes except Pythium (see literature cited by [2]). Preventatively, it inhibits zoospore release, zoospore motility, cystospore germination and direct germination of sporangia. Curatively, it stops mycelial growth, inhibits

**Competing interests:** The authors have declared that no competing interests exist.

lesion expansion and inhibits spore production. It shows translaminar and acropetally systemic movements [2–6]. Oxathiapiprolin is highly effective when applied to the foliage [2] the root system [3] or the seeds [4].

Resistance against OXPT was induced in *Phytophthora capsici* by UV irradiation [7,8]. These authors reported on three point-mutations that confer resistance to OXPT in *P. capsica* and *P. sojae*. Mutants with any of these mutations may have survival potential in the field [7].

According to FRAC (http://www.frac.info/), the resistance risk of OXTP is medium to high. Therefore, resistance management is required, including mixing with another fungicide with a different mode of action. Oxathiapiprolin mixtures (with azoxystrobin, mandipropamid or mefenoxam) were reported to be highly effective against *P. infestans* in tomato [5] and *Pseudoperonospora cubensis* in cucumber [6]. OXPT+mefenoxam was effective against mefenoxam-resistant isolates of these pathogens [5,6].

Benthiavalicarb belongs to the carboxylic acid amides (CAA) group of fungicides (FRAC code 40). CAA mode of action involves inhibition of cell wall synthesis of oomycetes by blocking the activity of cellulose synthase Ces3A [9]. Benthiavalicarb inhibits mycelia growth, zoosporangia germination cystospore germination, and sporulation of *Phytophthora infestans* at very low concentration [10]. The fungicidal and disease-controlling activities of benthiavalicarb are characterized by its preventive, curative, translaminar, systemic movement, residual activity and inhibitory activity toward lesion development [10].

Sensitivity monitoring studies over several years revealed that in populations of the late blight pathogen, *P. infestans*, all isolates were fully sensitive to CAA fungicides. However, resistant isolates occur in populations of the grape downy mildew pathogen *Plasmopara viticola* and the cucurbit downy mildew pathogen *P. cubensis* (FRAC, annual meeting 2018). With *P. infestans*, we were able to artificially mutate sporangia for stable resistance to the phenylamide fungicide mefenoxam but failed to select mutants with stable resistance to CAAs [11], suggesting a low risk of resistance developing in this pathogen in field populations against CAAs. No laboratory or field resistance to oxathiapirolin have been reported in *P. infestans*.

In a recent study [3] we demonstrated that a single application of oxathiapiprolin, benthiavalicarb, or their mixture (3+7, w/w) to the root system of nursery plants grown in multi-cell trays provided prolonged systemic protection against late blight and downy mildews in growth chambers and in the field. Soil application of 1mg active ingredient per plant provided durable protection for up to four weeks in tomato against late blight, cucumber against downy mildew and basil against downy mildew. Other researchers demonstrated efficacy of soil drench with oxathiapiprolin against *Phytophthora* black shank in tobacco [12] *Phytophthora* blight in pepper [13] and *Phytophthora* root rot in citrus [14].

These findings provided the conceptual basis for the developing of a new strategy to control late blight in potato by fungicidal soil application. The new fungicidal mixture Zorvec Endavia© by Corteva looked ideal for this purpose as it was proven highly effective when applied as a root treatment to tomato against late blight [5]. It is composed of two systemic fungicides, oxathiapiprolin and benthiavalicarb (3+7, w/w) that can protect each other from resistance development due to their different modes of action. The fact that no resistance occurs in *P. infestans* population against either fungicide in nature encouraged us in developing this strategy.

## Materials and methods

### Fungicides

Oxathiapiprolin (OXPT) 100 OD (oil dispersion) was a gift from DuPont, France. Mefenoxam (= MFX, 480 SL) and Benthiavalicarb (BENT) 98% technical grade were a gift from Syngenta Crop Protection, Switzerland. OXPT+BENT (oxathiapiprolin+benthiavalicarb, Zorvec

Endavia = ZE), was a gift from Agrochem Ltd (Petach Tikva, Israel). Infinito 68.75% SC (Fluopicolide 62.5 g/L + Propamocarb-HCl 625 g/L) was a gift from Lidor Chemicals Ltd (Petach Tikva, Israel). All fungicides (except BENT) were suspended in water and diluted to a series of 10-fold concentration suspensions from 0.01 to 1000 μg ai (active ingredient) per ml. BENT was dissolved in DMSO (100 mg/10 ml) and then suspended in water. For the dual mixtures, an indicated dose represents the combined doses of both ingredients.

## Field experiments

Four field experiments were conducted: three at Bar-Ilan University (BIU) Farm, Ramat Gan, Israel, and one at Kibbutz Magen, Western Negev, Israel

In Experiment 1 at BIU Farm, tubers *cv* Nicola were sown (1.10.2019) in 64 L polystyrene containers (80x40x20 cm) filled with peat: perlite (10:1, v/v) soil mixture, 5 tubers (50–60 mm in size) per container. Containers were placed a 50x6 m net house covered with white, 50 mesh, insect-proof plastic screens. At 7 weeks after sowing (18.11.2019), 1L fungicide suspension containing 12.5, 25 or 50 mg ai of a fungicide was applied to the soil surface of each container. Five fungicides were used: OXPT, BENT, ZE, MFX and Infinito. Control plants were treated with 1L of water. Three containers were used per fungicide per dose treatment. At 1h and 5h after soil drench, 1L of water was added to each container to facilitate the uptake of the fungicides by the root system. Thereafter, plants were drip irrigated 4 times a day with 0.5 L per container each time. NPK fertilizer (0.5%) was supplied twice a week.

In Experiment 2 at BIU farm, potato *cv* Sifra were sown in new containers on 26.12.2019 and the fungicides were applied to the soil surface as above at 8 weeks after sowing (19.2.2020). Three fungicides were used: OXPT, BENT and ZE. MFX and Infinito were excluded due to their inefficacy in the previous experiment.

Experiment 3 was performed to determine whether the fungicides applied in experiment 1 at BIU Farm on 18.11.2019 persisted in the soil so to provide protection against late light to the new potato crop in the subsequent season. Potato tubers *cv* Nicola were sown on 7.1.2020 in the same containers of Experiment 1, two days after tuber harvest. At 10 weeks after planting (15.3.2020), 118 days after soil treatment, the plants grown in the containers were inoculated with sporangia of *P. infestans* as described below.

Experiment 4 was done at Kibbutz Magen in the Western Negev. Four plots (4x12m each, ~240 plants per plot in 4 rows of 60 plants per row) in a commercial field of potato Sifra were used. At 10 weeks after planting (9.12.2019), plots were treated as follows: untreated control; spray with ZE at the recommended dose of 40 g ai/ha; spray with ZE of 150 g ai/h which was immediately washed down with overhead sprinkling irrigation equivalent to 20 mm rain; spray with ZE of 500 g ai/h which was immediately washed down with 20 mm rain. The two last treatments were considered as a soil drench treatment.

## Estimation of disease control in field-grown plants

At various time intervals after soil drench (in Experiment 1, at 1–20 days; in Experiment 2, at 4–16 days) leaves were detached from the control and the treated plants (1 compound leaf per plant) and brought to the laboratory for inoculation with *P. infestans*. Leaves were placed on a wet filter paper in sealed transparent boxes (60x30x10 cm), upper surface uppermost, and spray-inoculated with sporangial suspension ($5x10^3$ sporangia /ml) of *P. infestans* isolate 164. This isolate was collected in March 2016 from potato at Nirim, Western Negev, Israel. It is resistant to mefenoxam (MFX) and belongs to the 23_A1 genotype. The inoculated leaves were placed in a dew chamber at 18˚C overnight and then in a growth chamber at 20˚C, 14h/day. Percent leaf area blighted was visually estimated at 7 days post inoculation (dpi), unless

stated otherwise. At 21 and 25 days after soil drench in Experiment 1 and 2, respectively, all plants in the net-house were spray-inoculated with of *P. infestans* ($5x10^3$ sporangia /ml) isolate 164. Plants in Experiment 3 were inoculated on the same day as the plants in Experiment 2.

Plants in Experiment 4 were sampled at 10, 21 and 43 days after treatment; one compound leaf was sampled from every 6th plant in a row. Leaves were brought to the laboratory and inoculated with *P. infestans* as described above. Disease development was visually estimated (% infected leaf area) in the detached leaves at 6 or 7 dpi.

### Data analysis

Effective dose values (ED90, the dose of a fungicide(s) required to reduce % leaf area blighted by 90% relative to the control) were derived from log-probit regression curves using SPSS software. SF (synergy factor) was calculated using the Wadley formulae as described before [15]. t-test was performed using SPSS software to determine significant differences between means at $\alpha = 0.05$.

## Results

### Experiment 1: Efficacy of soil drench in cultivar Nicola

Results presented in Fig 1 show the response to inoculation in growth chambers of the leaves which were detached from control and fungicide-treated plants at 1, 3, 7, 9, 15 and 20 days after soil drench. Leaves that were detached at 1 day from plants treated with ZE produced significantly less disease compared to leaves that were taken from plants treated with BENT or OXPT, demonstrating rapid uptake and synergistic interaction (SF = 4.31) between OXPT and BENT, the components of ZE (Fig 1A). Leaves that were taken from plants treated with MFX or Infinito produced as much disease as leaves that were taken from control plants (Fig 1A). Efficacy of OXPT and BENT increased at ≥3 days after soil drench, indicating on enhanced translocation to the leaves, reaching a similar level of protection as ZE (Fig 1B). OXPT, and ZE, and to a lesser extent BENT, maintained high control efficacy in leaves that were detached at 7, 9, 15 and 20 days after soil drench (Fig 1C–1F). MFX and Infinito were totally ineffective. The appearance of the disease in artificially inoculated leaves detached at 7 days after soil drench is shown in Fig 2.

At 21 days after soil treatment, plants grown in all containers were spray-inoculated *in situ* with sporangial suspension of *P. infestans* isolate 164. Disease development as was visually estimated at various time intervals after inoculation is shown in Fig 3A–3F. OXPT, BENT and ZE were protective against the blight in a dose dependent manner all along the epidemic period (Fig 3A–3C and 3F). The higher was the dose applied the better was the protection against the blight, except for OXPT in which no significant difference was seen between the highest two doses. MFX and Infinito were ineffective (Fig 3D–3F). Fig 4 shows the appearance of the plants before inoculation (left panel) and at 13 days post inoculation (right panel).

### Experiment 2: Efficacy of soil drench in cultivar Sifra

Results presented in Fig 5 show the response to inoculation with *P. infestans* in growth chambers of the leaves which were detached from control and fungicide-treated plants at 4, 6, 12 and 16 days after soil drench. OXPT, BENT and ZE contributed a high and significant protection against the blight in all sampling days. High level of protection was already seen as early as 4 days after treatment (data for 1 and 3 days are unavailable due to technical reasons). At this time, ZE provided better protection compared to OXPT (Fig 5A), but at later sampling times, all fungicides performed in a similar effective manner (Fig 5B–5D).

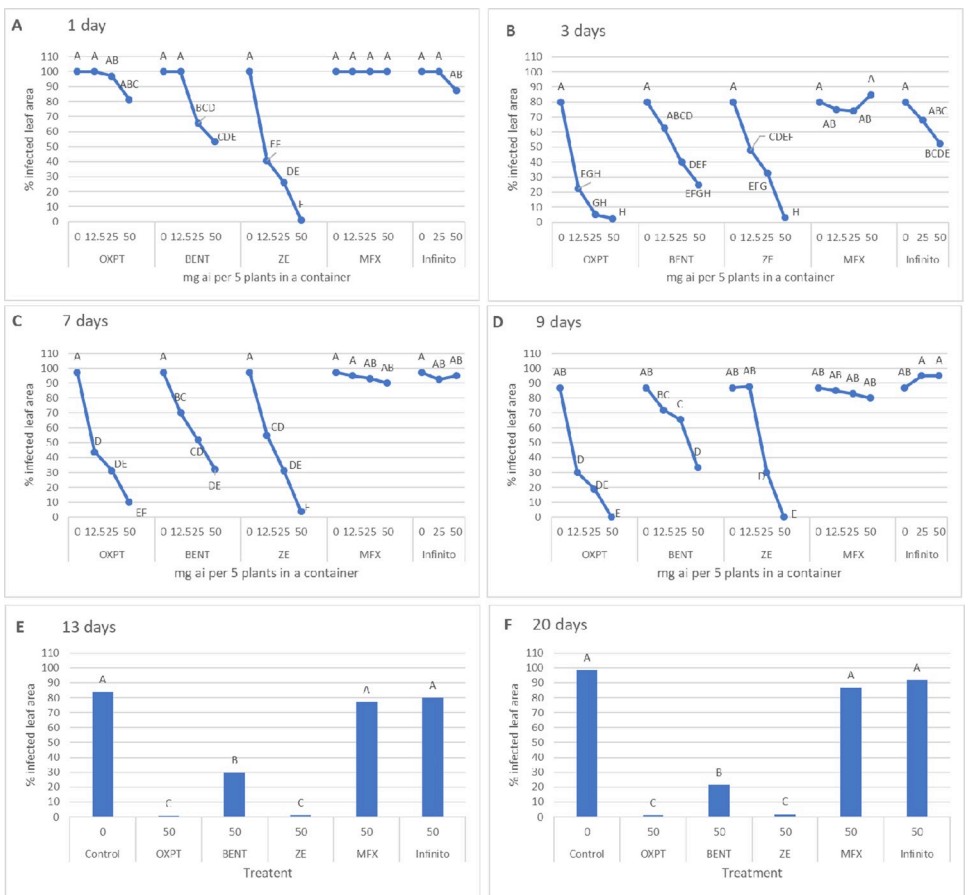

**Fig 1. Systemic protection of potato Nicola against late blight by each of five fungicides applied to the soil.** Six-weeks old potato plants growing a 64 L container were treated with a single soil-drench of 1L fungicide suspension containing 0, 12.5, 25 or 50 mg ai of a fungicide. Leaves (one compound leaf per plant) were detached at 1–20 days after treatment, inoculated with *Phytophthora infestans* in growth chambers and visually assessed for late blight development at 7 days post inoculation. Different letters on the curves or columns indicate on a significant different between means (n = 3) at α = 0.05.

At 25 days after soil drench, plants in all containers were spray inoculated *in situ* with sporangial suspension of *P. infestans* isolate 164. Late blight progress was visually estimated for 21 days after inoculation. The results are shown in Fig 6A–6D. All three fungicides were highly effective in protecting against the blight. At 50 mg ai per container, all fungicides were fully protective but at the lowest dose of 12.5 mg ai per container OXPT outperformed BENT and ZE. At 25 mg ai per container, ZE performed better that BENT (Fig 6A–6D). The appearance of the plants at 10 dpi (days post inoculation) is shown in Fig 7.

### Experiment 3: Persistence of fungicides in soil

The data presented in Fig 8A and 8E show that OXPT applied to the soil in the previous season at a dose of 25 or 50 mg ai per container provided in the present season excellent control of the disease for 21 days after inoculation, suggesting its persistence (or the persistence of its degradation products) in the treated soil for at least 139 days. BENT, MFX and Infinito provided no control at all (Fig 8B and 8E) whereas ZE provided partial control of the disease at 50 mg ai per container (Fig 8C and 8E). Fig 9 shows the appearance of the plants at 10 days post inoculation.

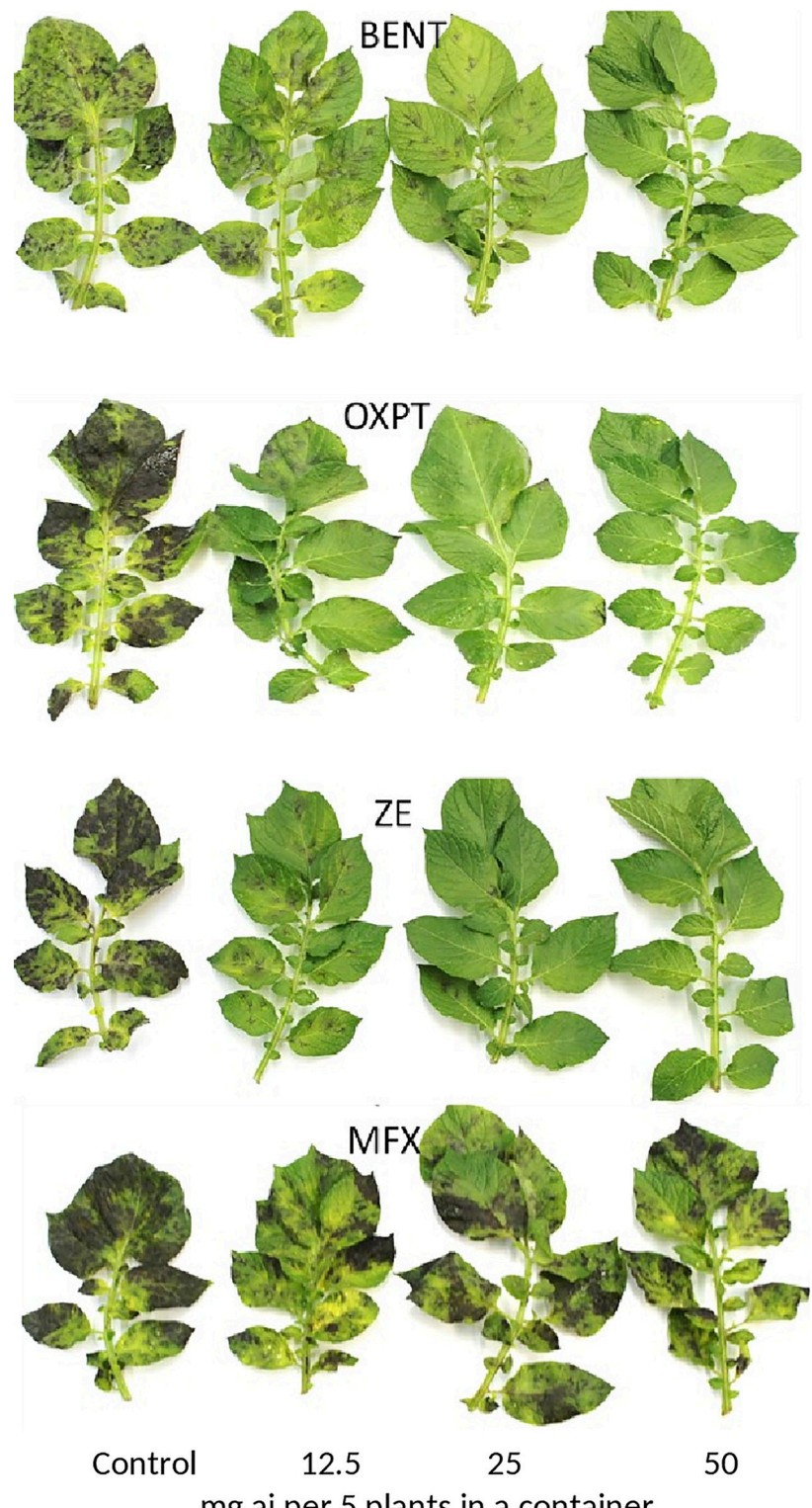

**Fig 2. Systemic protection of potato Nicola against late blight by 4 fungicides applied to the soil.** Six-weeks old field-grown plants were treated by a single soil-drench with 1L fungicide suspension containing 0, 12.5, 25 or 50 mg ai of the indicated fungicide. Leaves were detached at 7 days after soil drench and inoculated with *Phytophthora infestans* in a growth chamber at 20˚C. Photo was taken at 7 days post inoculation.

## Experiment 4: Comparative efficacy of ZE applied as a foliar spray or as a soil drench

The data in Table 1 show that all three treatments (a foliar spray and two soil drenches) were highly and significantly effective in controlling the blight in the leaves that were collected at 10 days after treatment. Control efficacy in leaves collected at 21 and 43 days after treatment was poor in the foliar spray treatment but very high in the two soil drench treatments, suggesting that the sprinkling irrigation applied after the spray was effective in enabling the wash down of ZE to the soil and facilitating its uptake to the foliage. The appearance of infected leaves (at 6 dpi) which were collected at 10 days after treatment is shown in Fig 10A–10D and of the leaves which were collected at 21 days after treatment is shown in Fig 10E–10G.

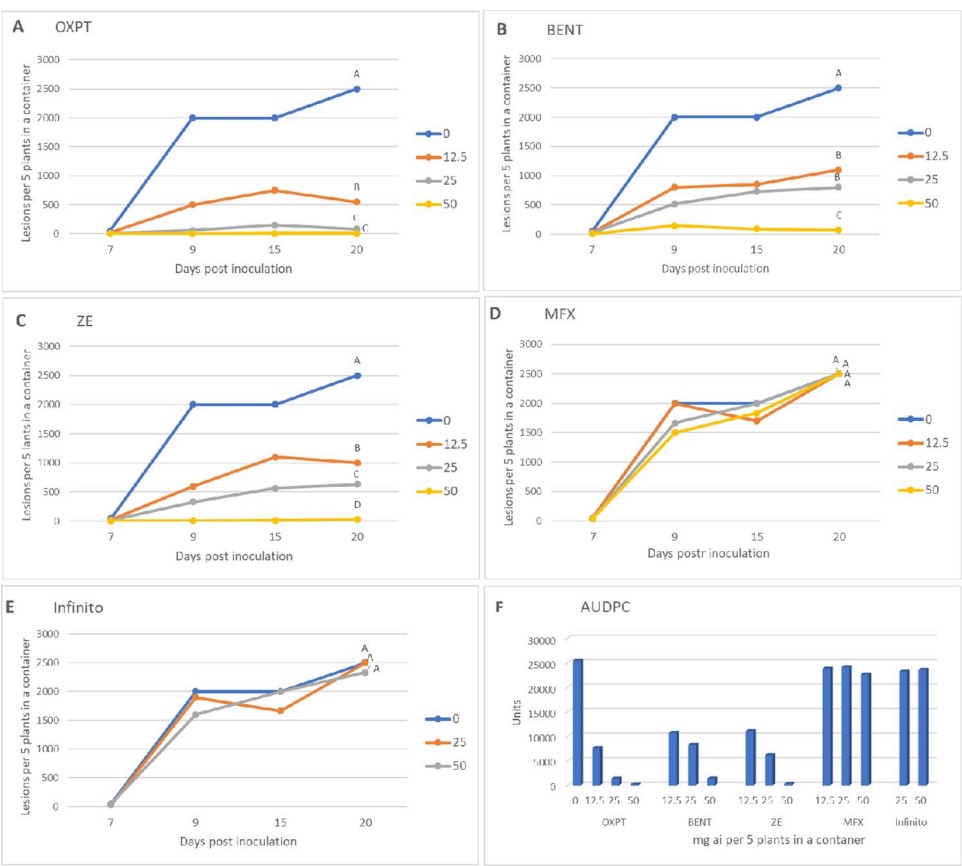

**Fig 3. Systemic protection of potato Nicola against late blight caused by an MFX-resistant isolate of *Phytophthora infestans* under field conditions.** Potato plants grown in 64 L containers were treated by a soil drench of 12.5, 25, or 50 mg ai fungicide suspension per container (n = 3). At 21 days after treatment the plants were inoculated with a sporangial suspension of isolate 164 of the pathogen. Percent leaf area occupied by late blight lesions was visually recorded at various time intervals after inoculation. Different letters on the curves indicate on significant differences between treatment at 20 dpi (t-test, α = 0.05). A- oxathiapiprolin. B- Benthiavalicarb. C- Zorvec Endavia. D- Infinito. E- Mefenoxam. F- Area under disease progress curve.

**Fig 4. Systemic protection of potato Nicola against late blight by 4 fungicides applied to the soil.** Six-weeks old field grown plants (left panel) were treated with fungicides via the soil. At 21 days after treatment plants were spray-inoculated with sporangial suspension of *Phytophthora infestans*. Photos of the infected plants (middle and right panels) were taken at 13 dpi. ZE, MFX, Infinito and OXPT, all applied at 50 mg ai per 5 plants in a container.

## Discussion

In a recent study [3] we showed that a single application of oxathiapiprolin, benthiavalicarb, or their mixture Zorvec Endavia (3+7, w/w) to the root system of nursery plants grown in multi-cell trays provided prolonged systemic protection against late blight and downy mildews in growth chambers and in field tests. Soil application of 1mg ai per plant provided durable protection of up to four weeks in tomato against late blight, cucumber against downy mildew and basil against downy mildew.

Others showed efficacy of soil drenches with oxathiapiprolin against black shank in tobacco [12], Phytophthora root rot in citrus [14] and *Phytophthora* blight in bell pepper [13]. No soil drench tests with the above fungicides were reported for potato.

The present research proves that a single application of oxathiapiprolin, benthiavalicarb or their mixture ZE to the soil in which potato plants are grown was highly protective against late blight caused by mefenoxam-resistant *P. infestans* in the field. ZE, composed of the two

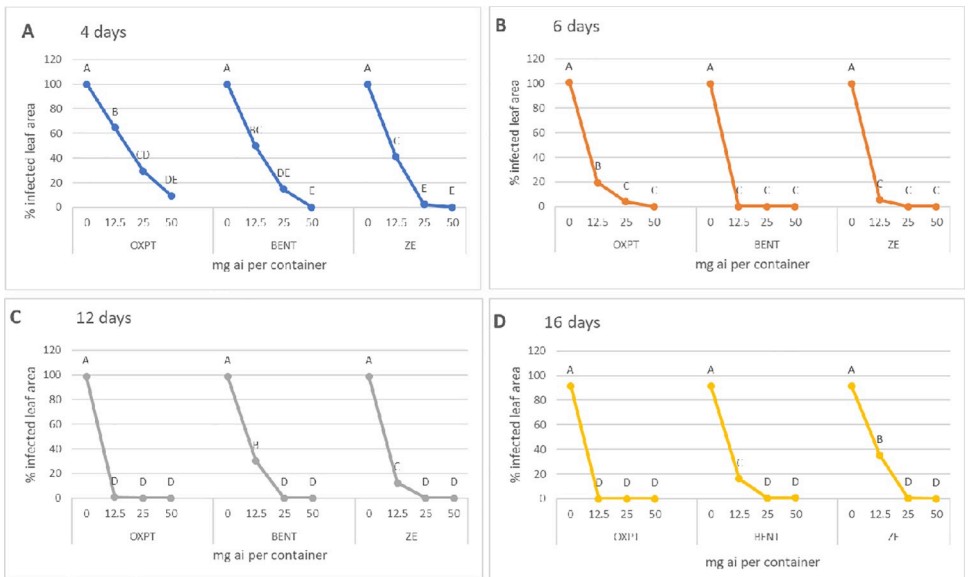

**Fig 5. Systemic protection of potato Sifra against late blight by each of three fungicides applied to the soil.** Five 8-weeks old potato plants growing a 64 L container were treated with a single soil-drench of 1L fungicide suspension containing 0, 12.5, 25 or 50 mg ai of a fungicide. Leaves (one compound leaf per plant) were detached at 4–16 days after treatment, inoculated with *Phytophthora infestans* in growth chambers and visually assessed for late blight development at 7 days post inoculation. Different letters on the curves indicate on a significant difference between means (n = 3) at α = 0.05.

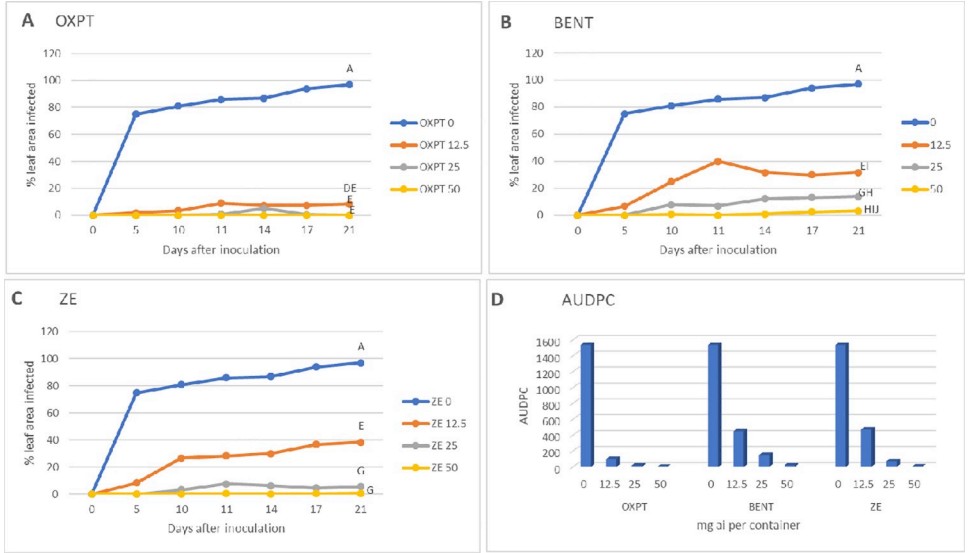

**Fig 6. Systemic protection of potato Sifra against late blight caused by an MFX-resistant isolate of *Phytophthora infestans* under field conditions.** Potato plants grown in 64 L containers were treated by a soil drench of 12.5, 25, or 50 mg ai fungicide suspension per container (n = 3). At 25 days after treatment the plants were inoculated with a sporangial suspension of isolate 164 of this pathogen. Percent leaf area occupied by late blight lesions was visually recorded at various time intervals after inoculation. Different letters on the curves indicate on significant differences between treatments (n = 3) at 21 dpi (t-test, α = 0.05). A- oxathiapiprolin. B- Benthiavalicarb. C- Zorvec Endavia. D- Area under disease progress curve.

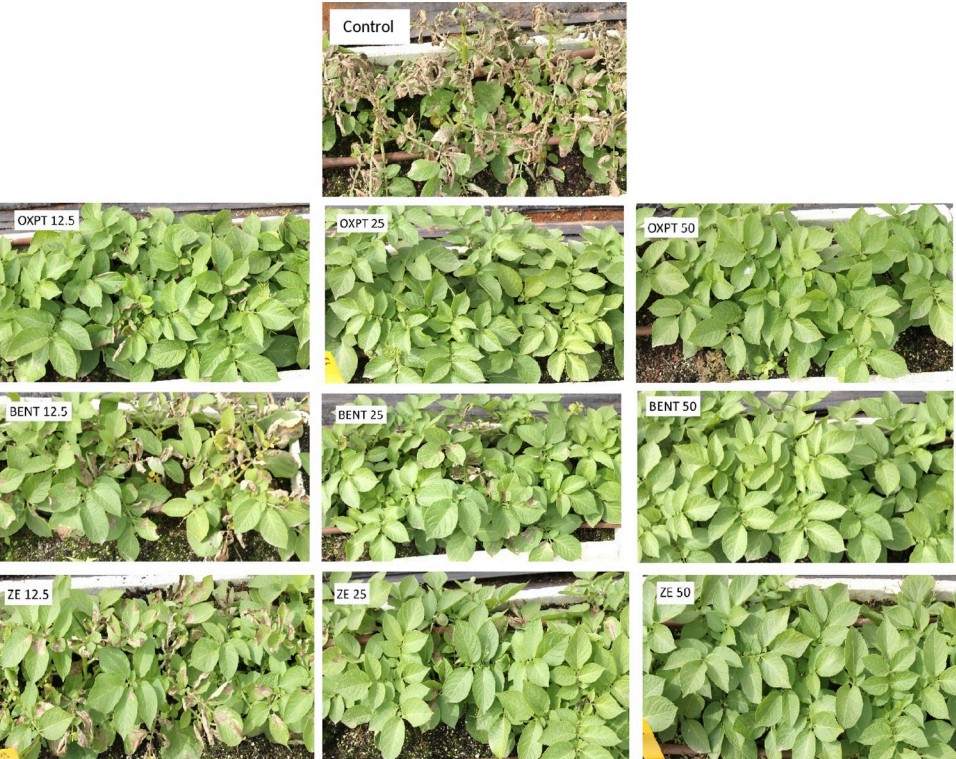

**Fig 7. Systemic protection of potato Sifra against late blight caused by an MFX-resistant isolate of *Phytophthora infestans* under field conditions.** Potato plants grown in 64 L containers were treated by a soil drench of 12.5, 25, or 50 mg ai fungicide suspension per container (n = 3). At 25 days after treatment the plants were inoculated with a sporangial suspension of isolate 164 of this pathogen. Photo was taken at 10 dpi.

systemic fungicides OXPT [2,3] and BENT [10], was the only commercial oxathiapiprolin mixture that with soil application to potted potato plants could provide effective systemic control of late blight caused by mefenoxam-resistant *P. infestans*. Dose response experiments showed ED90 values of 0.04, 7.52, 10.79, 12.18, 13.50 and 15.00 mg ai per pot/plant for OXPT +Benthiavalicarb, OXPT+Mefenoxam, OXPT+Azoxystrobin, OXPT+Mandipropamid, OXPT +Famoxadone, and OXPT+ Zoxamide, respectively (Cohen, *unpublished data*). In such experiments, a dose of 0.1 mg ai ZE applied to the root system of 12-leaf potted potato plant provided ~ 90% protection against the disease. Protection of the foliage was already detected at 1 day after root treatment, suggesting a rapid acropetal translocation of the compounds from the root. Qu et al [13] reported that when oxathiapiprolin was applied to the roots of bell pepper plants grown in hydroculture, the compound was detected in the second true leaf within 8 h and in the top new leaf 48 h after application to the root. In potted plants, translocation to the top new leaf was slower, within 3 days after application. Our data show that ZE reached the top leaf of 5 leaf tomato plants grown in soil, within one hour after soil drench (Cohen, *unpublished*).

Our data showed that ZE is a synergistic mixture, namely it provides a significant better disease control than the combined control efficacies of its two components [5,6]. OXPT was proved to act synergistically with mefenoxam in controlling *P. infestans* in tomato [5]. It also acts synergistically with mefenoxam and/or Bion in controlling downy mildew in sunflower [4].

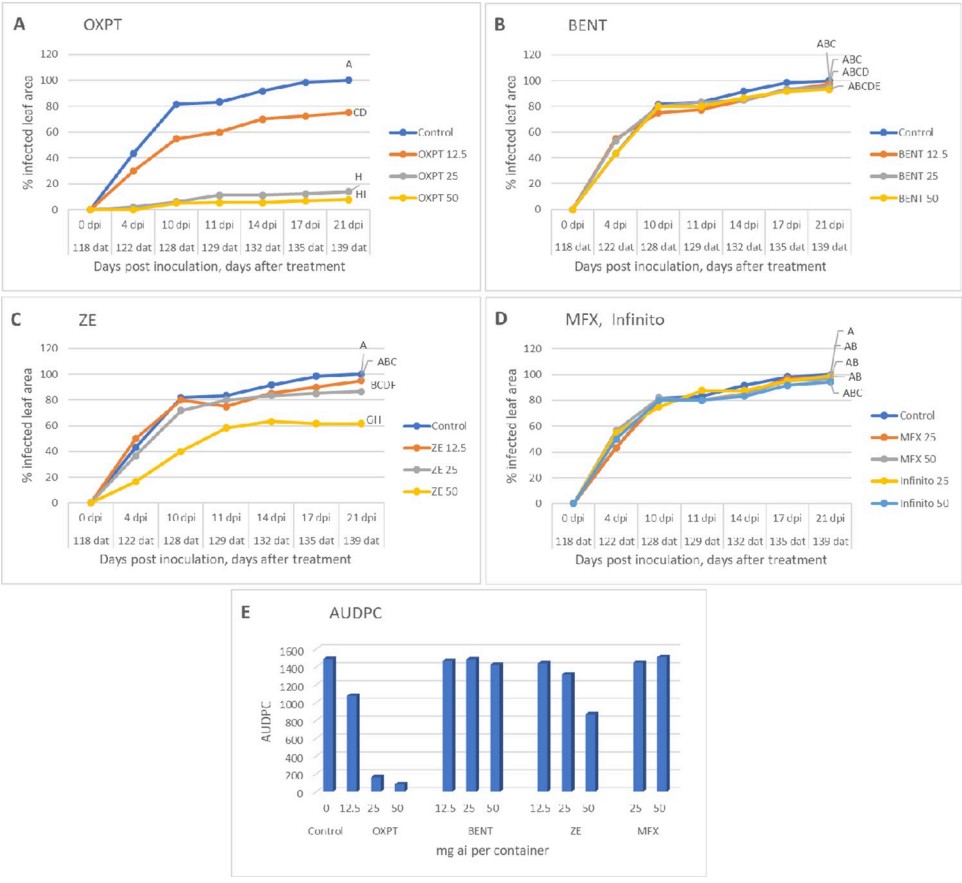

**Fig 8. Persistence of fungicidal activity in soil.** Soil in 64 L containers was drenched with each of five fungicides on 18.11.2019 and sown with potato Nicola on 7.1.2020, 118 days after treatment (dat). At about 10 weeks after sowing all plants were spray-inoculated with sporangial suspension of *Phytophthora infestans*. Disease progress was followed for 21 days after inoculation (139 dat) by recording the percent leaf area infected. A- oxathiapiprolin. B- Benthiavalicarb. C- Zorvec Endavia. D- Mefenoxam and Infinito. E- Area under disease progress curve. Different letters on the curves indicate on significant differences between treatments (n = 3) at 21 dpi (t-test, α = 0.05).

In the present field experiments, a single soil drench of ZE or OXPT provided detached leaves of potato with 20 days of high protection against *P. infestans*. BENT was moderately effective while Infinito (serving as a control) was ineffective. Mefenoxam was ineffective due to the resistance of *P. infestans* used to the fungicide. Treated plants in the field remained protected for additional 20 days after *in situ* inoculation, totaling about 6 weeks after root treatment. A similar long protective period *P. infestans* was reported in tomato [3].

Interestingly, the treated soil in the field remained protective against late blight even during the subsequent potato crop season, for more than 139 days after drench. According to PPDB: Pesticide Pesticides Properties DataBase, University of Hertfordshire (https://sitem.herts.ac.uk/aeru/ppdb/en/Reports/2618.htm#none), oxathiapiprolin is moderately persistent in soil with DT50 range of 31.5–138.5 days. Oxathiapiprolin degrades in soil into 4 metabolites of which 1-(2-(4-(4-(5-(2,6-difluorophenyl)-4,5-dihydro-3-isoxazolyl)-2-thiazolyl)-1-piperidinyl)-2-oxoethyl)-3-(trifluoromethyl)-1H-pyrazole-5-carboxylic acid (code IN-RAB06) is major (Estimated maximum occurrence fraction = 0.135). In a study performed by DuPont [16], when [14C] oxathiapiprolin was applied to potatoes foliage, oxathiapiprolin was the major residue component in the foliage at the end of the season, accounting for 25–59% TRR.

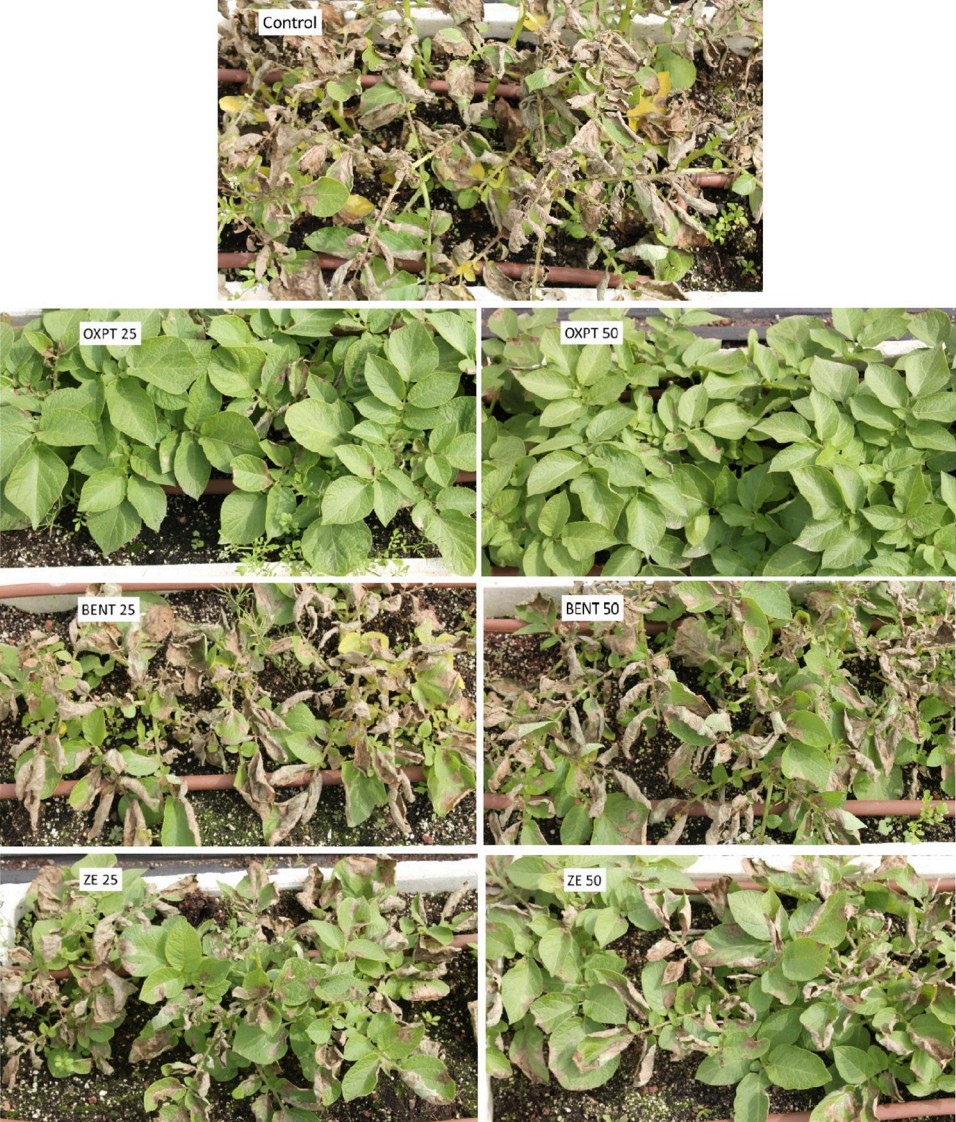

**Fig 9. Persistence of fungicidal activity in soil.** The appearance of the infected plants at 10 days after inoculation, 128 days after soil drench with OXPT, BENT or ZE at 25 or 50 mg ai per 5 plants in a container.

**Table 1. Comparative efficacy of ZE applied to potato crops in the field at Kibbutz Magen, Israel by foliar spray or soil drench.**

|  | % infected leaf area | | |
| --- | --- | --- | --- |
|  | Days after soil drench | | |
| **ZE Applied** | **10** | **21** | **43** |
| **Control untreated** | 96 a | 100 a | 100 a |
| **Foliar spray, 40 g ai/ha** | 8.9 b | 81.4 ab | 74.3 ab |
| **Soil drench, 150 g ai/ha** | 1.5 c | 23.2 c | 30.7 c |
| **Soil drench, 500 g ai/ha** | 0 c | 0 d | 1.8 d |

Ten weeks old potato plants (240 plants in 48 m$^2$ plot) were sprayed once or sprayed and immediately irrigated by overhead sprinklers with 20 mm of rain. Compound leaves (n = 15–20) were detached at 10, 21 and 43 days after treatment and inoculated in growth chambers with sporangial suspension of *Phytophthora infestans* isolate 164. Percent leaf area infected was visually estimated in each leaf at 6 dpi. Values followed by different letters in columns indicate on a significant difference between treatments (t-test, α = 0.05).

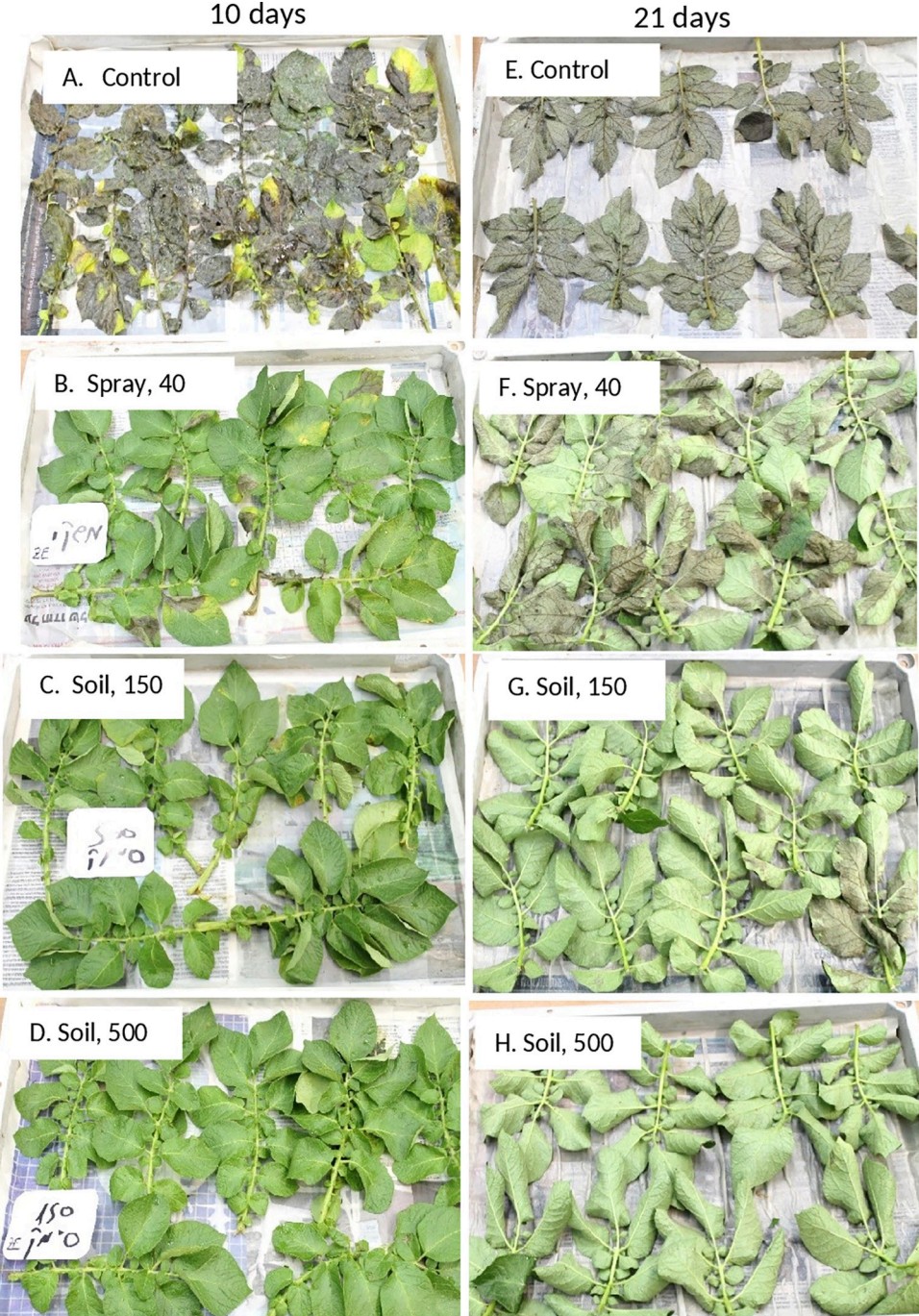

**Fig 10. Compared efficacy of ZE applied as a foliar spray or as a soil drench to potato Sifra in the field against artificial inoculation with *Phytophthora infestans*.** Ten weeks old plants were sprayed with 40 g ai/ha or treated with a soil drench of 150 or 500 g ai/ha. Leaves were detached at 10 and 21 days after treatment and inoculated with sporangial suspension of the pathogen in growth chambers at 20˚C. Photos was taken at 7 dpi.

A range of minor metabolites were individually present at no greater than 8% TRR. However, when [$^{14}$C] oxathiapiprolin was applied to the soil and foliage and tubers were sampled at 37 and 72 days after treatment, oxathiapiprolin was not a major residue, present at < 10% TRR

and < 0.005 mg/kg in tubers and foliage. Rather, IN-WR791 (5-Methyl-3-(trifluoromethyl)-1H-pyrazole-1-acetic acid) was the major component identified in mature tubers (25% TRR and 0.003 mg eq/kg). Also present at levels of 0.001–0.002 mg eq/kg were IN-E8S725 (trifluoromethyl)-1H-pyrazole-3-carboxylic acid) (14% TRR) and IN-RZB205 (hydroxymethyl)-3-(trifluoromethyl)-1H-pyrazole-1-acetic acid (12% TRR). The principal components identified in the foliage (11–13% TRR, up to 0.015 mg eq/kg) were IN-RZB215-(hydroxymethyl)-3-(trifluoromethyl)-1H-pyrazole-1- acetamide), IN-RZD74 (3-(trifluoromethyl)-1H-pyrazole-5-methanol) and IN-RZB20 (5-(hydroxymethyl)-3-(trifluoromethyl)-1H-pyrazole-1-acetic acid) [16]. These data suggest that pyrazole metabolites of oxathiapiprolin may be responsible for the protection of potato foliage from late blight after soil application of oxathiapiprolin.

Soil application of systemic pesticides may save farmers with labor and costs. However, it also exposes the pathogen to a prolonged selection pressure which may result with resistance of the pathogen to the pesticide. OXPT is a medium to high risk fungicide (FRAC code list 2014) prone to "breakdown" due to the appearance of resistant variants of oomycetes [1,7,8]. One important approach to minimize the resistance risk is to mix the vulnerable fungicide with another fungicide with a different mode action. Indeed, ZE is a mixture that gives a strong anti-resistance strategy. The one component, OXPT, is a piperidinyl-thiazole-isoxazoline fungicide (FRAC code U15), inhibitor of sterol binding proteins and the other, BENT (FRAC code 40), is a carboxylic acid amide fungicide inhibitor of cellulose synthase. BENT is a low to medium risk fungicide with resistance known in *Plasmopara viticola* but not in *P. infestans* (Frac list 2018, [11]). As both fungicides are systemic [3–6,13,14] they should mutually protect each other against resistance development even during a prolonged exposure period in the field. Experimental field data are required to validate this hypothesis

In conclusion, two major findings are here reported:

i. ZE is the only oxathiapiprolin mixture that can systemically protect potato foliage against late blight following a soil application.

ii. A single soil application of ZE provides full season protection against late blight caused by mefenoxam resistant *P. infestans*.

## Acknowledgments

We thank Uri Zig of YAHAM, Israel, for cooperating with Experiment 4.

## Author Contributions

**Conceptualization:** Yigal Cohen.

**Data curation:** Avia E. Rubin.

**Investigation:** Avia E. Rubin.

**Project administration:** Yigal Cohen.

**Supervision:** Yigal Cohen.

**Validation:** Yigal Cohen.

**Writing – review & editing:** Yigal Cohen.

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
