## [Decision Letter · Decision Letter 0]

31 Jul 2020

PONE-D-20-11462

A new strategy for durable control of late blight in potato by a single soil application of an oxathiapiprolin mixture in early season

PLOS ONE

Dear Dr. cohen,

Thank you for submitting your manuscript to PLOS ONE. After careful consideration, we feel that it has merit but does not fully meet PLOS ONE’s publication criteria as it currently stands. Therefore, we invite you to submit a revised version of the manuscript that addresses the points raised during the review process.

ACADEMIC EDITOR:

We look forward to receiving your revised manuscript.

Kind regards,

Vijay Kumar

Academic Editor

PLOS ONE

Journal Requirements:

Reviewers' comments:

Reviewer's Responses to Questions

**Comments to the Author**

1. Is the manuscript technically sound, and do the data support the conclusions?

Reviewer #1: Yes

2. Has the statistical analysis been performed appropriately and rigorously? 

Reviewer #1: Yes

3. Have the authors made all data underlying the findings in their manuscript fully available?

Reviewer #1: Yes

4. Is the manuscript presented in an intelligible fashion and written in standard English?

Reviewer #1: Yes

5. Review Comments to the Author

Reviewer #1: Title

The title of the article is appropriate and accurately reflect the content of the paper.

Abstract

It represent the article nicely.

Introduction

• Phytophthora infestans should be in italics. Change it throughout the manuscript.

• Hyperlink the mentioned web link: http://www.frac.info/

• References in the manuscript should start from [1] in an increasing number according to the text flow.

• [10;11] should be [10,11]

• [5;6] should be [5,6]

• Plasmopara viticola should be in italics.

Materials and Methods

Well explained in detail.

Results:

Ok, results are significant and scientifically sound good

Conclusion

Well written.

6. PLOS authors have the option to publish the peer review history of their article (what does this mean?). If published, this will include your full peer review and any attached files.

Reviewer #1: No

---

## [Author Response · Author response to Decision Letter 0]

5 Aug 2020

All comments by the reviewers were adopted

---

## [Decision Letter · Decision Letter 1]

11 Aug 2020

A new strategy for durable control of late blight in potato by a single soil application of an oxathiapiprolin mixture in early season

PONE-D-20-11462R1

Dear Dr. cohen,

We’re pleased to inform you that your manuscript has been judged scientifically suitable for publication and will be formally accepted for publication once it meets all outstanding technical requirements.

Kind regards,

Vijay Kumar

Academic Editor

PLOS ONE

Additional Editor Comments (optional):

Reviewers' comments:

Reviewer's Responses to Questions

**Comments to the Author**

1. If the authors have adequately addressed your comments raised in a previous round of review and you feel that this manuscript is now acceptable for publication, you may indicate that here to bypass the “Comments to the Author” section, enter your conflict of interest statement in the “Confidential to Editor” section, and submit your "Accept" recommendation.

Reviewer #1: All comments have been addressed

2. Is the manuscript technically sound, and do the data support the conclusions?

Reviewer #1: Yes

3. Has the statistical analysis been performed appropriately and rigorously? 

Reviewer #1: Yes

4. Have the authors made all data underlying the findings in their manuscript fully available?

Reviewer #1: Yes

5. Is the manuscript presented in an intelligible fashion and written in standard English?

Reviewer #1: Yes

6. Review Comments to the Author

Reviewer #1: I have gone through the manuscript and Found that all the comments have been addressed by authors. Now this manuscript is recommended for final publication

7. PLOS authors have the option to publish the peer review history of their article (what does this mean?). If published, this will include your full peer review and any attached files.

Reviewer #1: No

---

## [Editor Report · Acceptance letter]

12 Aug 2020

PONE-D-20-11462R1 

A new strategy for durable control of late blight in potato by a single soil application of an oxathiapiprolin mixture in early season 

Dear Dr. Cohen:

I'm pleased to inform you that your manuscript has been deemed suitable for publication in PLOS ONE. Congratulations! Your manuscript is now with our production department. 

Kind regards, 

on behalf of

Dr. Vijay Kumar 

Academic Editor

PLOS ONE